# Antibiotics as a Stressing Factor Triggering the Harboring of *Helicobacter pylori* J99 within *Candida albicans* ATCC10231

**DOI:** 10.3390/pathogens10030382

**Published:** 2021-03-23

**Authors:** Kimberly Sánchez-Alonzo, Libnny Belmar, Cristian Parra-Sepúlveda, Humberto Bernasconi, Víctor L. Campos, Carlos T. Smith, Katia Sáez, Apolinaria García-Cancino

**Affiliations:** 1Laboratory of Bacterial Pathogenicity, Department of Microbiology, Faculty of Biological Sciences, Universidad de Concepción, Concepción 4070386, Chile; kimsanchez@udec.cl (K.S.-A.); lbelmar@udec.cl (L.B.); cparras@udec.cl (C.P.-S.); csmith@udec.cl (C.T.S.); 2Laboratorio Pasteur, Concepción 4030000, Chile; hbernasconi@lpasteur.cl; 3Laboratory of Environmental Microbiology, Department of Microbiology, Faculty of Biological Sciences, Universidad de Concepción, Concepción 4070386, Chile; vcamapos@udec.cl; 4Department of Statistics, Faculty of Physical and Mathematical Sciences, Universidad de Concepción, Concepción 4070386, Chile; ksaez@udec.cl

**Keywords:** *Helicobacter pylori*, *Candida albicans*, antibiotic, internalization, eradication failure

## Abstract

First-line treatment for *Helicobacter pylori* includes amoxicillin and clarithromycin or metronidazole plus a proton pump inhibitor. Treatment failure is associated with antibiotic resistance and possibly also with internalization of *H. pylori* into eukaryotic cells, such as yeasts. Factors triggering the entry of *H. pylori* into yeast are poorly understood. Therefore, the aim of this study was to evaluate whether clarithromycin or amoxicillin trigger the entry of *H. pylori* into *C. albicans* cells. Methods: *H. pylori* J99 and *C. albicans* ATCC 10231 were co-cultured in the presence of subinhibitory concentrations of amoxicillin and clarithromycin as stressors. Bacterial-bearing yeasts were observed by fresh examination. The viability of bacteria within yeasts was evaluated, confirming the entry of bacteria into *Candida*, amplifying, by PCR, the *H. pylori*
*16S rRNA* gene in total yeast DNA. Results: Amoxicillin significantly increased the entry of *H. pylori* into *C. albicans* compared to the control. Conclusion: the internalization of *H. pylori* into *C. albicans* in the presence of antibiotics is dependent on the type of antibiotic used, and it suggests that a therapy including amoxicillin may stimulate the entry of the bacterium into *Candida*, thus negatively affecting the success of the treatment.

## 1. Introduction

*Helicobacter pylori* is a bacterium that colonizes the human stomach, causing pathologies such as gastritis, peptic ulcer and malignancies including gastric adenocarcinoma and B cell lymphomas of the gastric mucosa-associated lymphoid tissue [1], which is why it is considered a type 1 carcinogen by the International Agency for Research on Cancer [2]. The prevalence of *H. pylori* has varied within the last twenty years in different regions of the world, being higher in Africa and Asia. Its prevalence of infection in South America is led by Chile (74.6%), Ecuador (72.2%) and Brazil (71.2%) [3]. Taking into consideration that *H. pylori* has been associated with the development of cancer, the study of this pathogen’s antibiotic resistance and its prevalence has been relevant since its discovery. Some of the aims of these studies have been the search for tools to elucidate the transmission mechanisms of *H. pylori* infection and to create treatment schemes using antibiotics, which will allow js to eradicate the infection by this pathogen [1,4,5,6,7,8]. 

Over the last twenty years, the treatment recommended to eradicate this bacterium has been the standard triple therapy, including an inhibitor of the proton pump or ranitidine–bismuth–citrate associated with clarithromycin (CLT) and amoxicillin (AMX) or metronidazole (MTZ). Nevertheless, the increasing resistance of *H. pylori* to the antibiotics used, mainly CLT, has reduced the success of this treatment scheme. Therefore, second-line therapies such as the sequential or quadruple ones have been used. Even more, third- or fourth-line therapies have been used when the first two have failed [9,10,11,12]. The main challenge in the treatment of *H. pylori* is its resistance to the treatment with antibiotics, which is alarmingly high in certain geographical regions for antibiotics such as CLT, MTZ and levofloxacin, and is attributed to their abuse [8,13,14]. Therefore, *H. pylori* strains resistant to CLT have been defined as a high priority by the WHO to search for and develop new drugs to help in their treatment [15].

Although treatments have lost their effectiveness, mainly due to the emergence of CLT-resistant strains, there are other factors that also contribute to their failure. In this respect, it has been shown that certain drugs used to eradicate this bacterium, such as AMX, CLT or MTZ, might act as stressing factors for *H. pylori*, triggering morphological and physiological adaptations, such as the coccoid morphology [16,17]. The coccoid morphology may be clinically associated with resistance, transmission of the infection and its recurrence after antibiotic treatment [18]. Moreover, within the last years, sufficient evidence has been gathered indicating that *H. pylori* may in vivo and in vitro be harbored within several eukaryotic cells, such as human gastric epithelial cells, immune cells, amoebas and *Candida* yeasts. These findings suggest that this bacterium may protect itself from challenges that threaten its viability, such as changes in environmental pH or eluding antibiotic treatments to eradicate it [7,19,20,21,22]

Although not much evidence exists on the factors promoting the entry of *H. pylori* into yeast cells [23], it has been demonstrated that the presence of antibiotics MTZ and CLT in the culture medium significantly increased the entry of *H. pylori* [7]. This supports the results of Lai and coworkers [23] co-culturing GES-1 cells and *H. pylori* in the presence of AMX, furazolidone, CLT, MTZ and levofloxacin, and their obtained results were similar to those of previous authors. Thus, they concluded that *H. pylori* protects itself from antibiotics harboring within other cells, evading the eradication of *H. pylori* infection.

Recently, our research group demonstrated that acidic pH, as a stress factor, induces the entry of *H. pylori* into *Candida albicans* cells, where the bacterium may find a shelter against unfavorable pH conditions within the yeast cells [24], a yeast that colonizes the gastrointestinal tract of nearly 40% of healthy humans [25,26]. Similarly, certain antibiotics may also trigger the harboring of *H. pylori* within *C. albicans;* therefore, the *H. pylori* and *C. albicans* interaction may be an example of the bacteria–yeast relationship, which may cause the failure of anti-*H. pylori* treatments. Thus, the aim of the present work was to evaluate if the antibiotics used in the triple therapy against *H. pylori* might stimulate, acting as stressing agents, the harboring of *H. pylori* cells within the yeast *C. albicans.*

## 2. Results

### 2.1. Growth Curves of H. pylori J99 (ATCC 700824) Strain and C. albicans ATCC10231 Strain

The growth curves of *H. pylori* J99 and *C. albicans* ATCC10231 strains cultured in Brucella broth supplemented with 10% fetal calf serum (BB-FCS) under aerobic conditions are shown in Figure 1. Based on these results, the co-cultures of these two microorganisms for the following experiments were prepared using 20 h pure cultures of *H. pylori* J99 and 10 h pure cultures of *C. albicans* ATCC 10231 strains. 

### 2.2. H. pylori J99-C. albicans ATCC 10231 Co-Culture Assay in the Presence of a Subinhibitory Concentration (¼ Minimal Inhibitory Concentration MIC = 0.0075 μg mL^−1^) of AMX or CLT

Wet mountings, observed under an optical microscope, showed the presence of bacteria-like bodies (BLBs) within yeasts and hyphae of *C. albicans*. Actively moving BLBs were observed within almost all hyphae observed (Figure 2). In the control co-culture, that is, in the absence of antibiotics, 25% and 34% of hyphae showed the presence of intracellular BLBs at 3 h and 12 h, respectively. The latter percentage remained unchanged until the end of the 48 h of culture. The co-culture in the presence of ¼ MIC CLT showed percentages of yeast cells containing BLBs not significantly different from those of the control co-culture (21% and 33% at times 3 h and 12 h, respectively) (Figure 3). On the contrary, in the presence of ¼ MIC AMX, the results were significantly different from those of the control and CLT containing co-cultures, showing the presence of yeasts containing BLBs from time zero on, and the percentages grew rapidly to reach 80% after 48 h of incubation (Figure 3).

### 2.3. Cell Viability Assay 

Observations of co-cultures using fluorescence microscopy after staining with the LIVE/DEAD BacLight Bacterial Viability kit showed that both the yeast cells and the BLBs they contained were viable, as indicated by the green fluorescence. It was also possible to observe the movement of BLBs within yeast cells, as shown by time-lapse photography in Figure 4.

### 2.4. Detection of H. pylori Genes in Total DNA Extracted from C. albicans Cells

The presence of intrayeast *H. pylori* was confirmed, amplifying, by PCR, the 16S rDNA gene of *H. pylori* in total *C. albicans* DNA of yeast cells previously co-cultured with the bacterium in the presence of antibiotic (¼ MIC AMX or ¼ MIC CLT). After agarose gel electrophoresis, the presence of an amplicon having the expected size in all samples of yeasts co-cultured with the bacterium and in the positive control (pure *H. pylori*) was revealed. The amplicon was not detected in the negative control (pure *C. albicans*) (Figure 5).

## 3. Discussion

In order to obtain co-cultures containing the higher possible number of viable cells of both microorganisms, the first task was to determine the growth curves for each one of them. The growth curve obtained in this work for *H. pylori* J99 strain was similar to that reported for the same strain by Baltrus and Guillemin [27] cultured under similar conditions. Other strains of this species also showed a similar growth when cultured in other culture media [28,29,30]. The growth of *C. albicans* ATCC 10231 in BB-FCS was not different from that reported when cultured in yeast extract peptone-dextrose (YPD) broth or Sabouraud broth, even at different temperatures [31,32,33].

The second task was to determine the MIC of AMX and CLT for both microorganisms, in order to know what concentration was stressful but not inhibitory for *H. pylori*; in addition, to know whether the antibiotics used had any effect on the growth of the yeast. The MIC of both antibiotics for *H. pylori* was 0.03 μg mL^−1^. Therefore, in accordance with the reference values reported by EUCAST [34], the *H. pylori* J99 strain was sensitive to AMX (resistant > 0.125 μg mL^−1^) and also to CLT (resistant > 0.5 μg mL^−1^). In our internalization assays, we used a sub-MIC concentration, specifically 0.0075 μg mL^−1^, equivalent to ¼ MIC. This concentration was chosen because studies reported for *H. pylori* using different antibiotics have used this same fraction of the MIC [35,36]. 

It has been postulated that the internalization of bacteria within eukaryotic microorganisms could represent a mechanism to persist under unfavorable conditions [24,37,38]. Therefore, the harboring of *H. pylori* within nearly 30% of *C. albicans* cells after 48 h of co-culture in the absence of antibiotics can be attributed to the basal stress caused by factors such as the transfer from solid media to broth, extended periods of incubation and the accumulation of metabolites. In the present work, other factors, such as deprivation of nutrients and pH changes, extensively discussed in [39,40], were kept under control. To the above-mentioned factors, the additional stress caused by the presence of antibiotic at ¼ MIC concentration was added as a possible triggering factor for the internalization of *H. pylori* within *C. albicans*. Results showed that, when compared to the control, AMX caused a significant increase in the internalization of this bacterium into yeast cells, while CLT did not. These results suggest that CLT does not fulfill the necessary requirements to trigger the harboring of *H. pylori* within *C. albicans* cells. Studies on the individual effect of antibiotics, such as AMX or CLT, on *H. pylori* have reported that they have similar effects on the morphological changes, biofilm formation and genetic modifications in this species [41,42]. Nevertheless, the significantly different percentage of *H. pylori* harboring yeasts suggests that the mechanism of action of the antibiotic may be playing an important role in the pressure exerted on *H. pylori* to trigger its internalization.

AMX is a β-lactam antibiotic that irreversibly binds to penicillin-binding proteins (PBP), having a preference for PBP2, interrupting the synthesis of peptidoglycan [43]. PBP2 is a D,D-transpeptidase closely associated with bacterial morphology, mainly to the lateral peptidoglycan synthesis, which generates the bacillar morphology. Therefore, the impairment of the function of PBP2 brings, consequently, the generation of the coccoid morphology in bacteria such as *H. pylori* and *E. coli* [18,44,45,46]. Thus, AMX, having a direct effect on the stability of the structure of *H. pylori*, becomes a source of stress, which could lead to *H. pylori* generating more quickly survival strategies, such as entering into yeast cells to protect itself from the effect of AMX. These considerations may justify the higher percentage of *H. pylori*-bearing yeasts in the co-cultures where this antibiotic (AMX) was used as a stressor.

On the contrary, CLT, being a member of the macrolide family of antibiotics, reversibly binds to 23S rRNA of the 50S subunit of the ribosome, the region neighboring the peptidyl transferase, an enzyme catalyzing the binding of the peptide during elongation. Hence, the binding of CLT interrupts protein synthesis [47]; therefore, it affects the viability of bacteria. Sub-inhibitory CLT concentrations cause several effects in other bacterial species, such as a decreased adherence of *Staphylococcus aureus* to cells of the oral mucosa [48]. On *Pseudomonas aeruginosa*, sub-inhibitory CLT concentrations cause a low rate of synthesis of various virulence factors, such as elastase, DNase, proteases and lecithinase, resulting in a decreased motility, suppression of twitching and inhibition of the biofilm matrix and proteases synthesis by bacteria [49,50,51]. In *H. pylori,* sub-inhibitory concentrations of CLT have been shown to cause the shift from the helical to the coccoid morphology and to increase biofilm formation [41,52,53] mechanisms used by *H. pylori* to protect itself from environmental stress, contributing to the fact that the entry of yeasts is not the only protection mechanism that this pathogen has to avoid the damage generated by the presence of CLT. This would support the results obtained in this work, where it was observed that ¼ MIC of CLT did not show any effect on the entry of bacteria into *C. albicans* when compared with the results obtained for AMX and the control. This may be explained by the reversibility of the action mechanism of this antibiotic at ¼ MIC concentration, minimizing the antibiotic pressure as a source of stress promoting the entry of *H. pylori* into eukaryotic cells. CLT pressure could be greater if the concentration of the antibiotic is increased. In fact, higher concentrations of CLT cause an increase in coccoid structures as a response to stress [52].

It is noteworthy that this study showed that *H. pylori* also enters into the yeast cells in the absence of an apparent stressing factor but that the number of yeast cells harboring BLBs increased in the presence of AMX. On the other hand, the cell viability assay (Figure 4) shows that *H. pylori* remains viable within the yeast because, as mentioned in Materials and Methods, yeasts harboring BLBs were reseeded six times in Sabouraud agar medium containing chloramphenicol antibiotic before the assay to evaluate viability. This assay was complemented with the amplification of the *H. pylori rRNA 16S* gene from the total DNA extracted from *C. albicans* cells in which BLBs were observed (Figure 5). 

It has been proposed that the presence of bacteria within eukaryotic microorganisms is a strategy to persist in the environment and even a strategy for dissemination [22,37,54,55,56]. The present study provides the first evidence of the in vitro entry of *H. pylori* cells into *C. albicans* cells promoted by the presence of antibiotics, mainly AMX, used for the treatment of this bacterium. This finding may reveal a protecting mechanism of *H. pylori* to cope with the presence of antibiotics used to treat it, making it difficult the eradicate it in clinical practice. It becomes necessary to pursue, in this line of research, the consequences of the harboring of *H. pylori* within eukaryotic cells and to improve our knowledge regarding the factors favoring this symbiosis. 

## 4. Materials and Methods

### 4.1. Strains and Culture Conditions

The strains used in this study included *H. pylori* ATCC 700824, also regularly known as J99 straina denomination also used in this manuscript, and *C. albicans* ATCC 10231. Both strains are maintained at the culture collection of the Laboratory of Bacterial Pathogenicity, Department of Microbiology, University of Concepcion, Concepcion, Chile. All the procedures described below were carried out aseptically.

*H. pylori* was cultured on plates containing Columbia agar (OXOID, Basingstoke, United Kingdom) supplemented with 5% horse blood Plates were incubated at 37 °C for 36 h under microaerobic conditions (10% CO_2_ and 5% oxygen) in a 3429-model incubator (Thermo Scientific, Waltham, MA, USA). *C. albicans* was cultured on plates containing Sabouraud dextrose agar (Merck, Darmstadt, Germany) supplemented with chloramphenicol, following the instructions of the manufacturer (OXOID, Basingstoke, United Kingdom). Incubation was carried out at 37 °C during 24 h under aerobic conditions in a ZDP-2160 model incubator (ZHICHENG, Shanghai, China). The purity of *H. pylori* cultures was verified by Gram staining and urease production using the rapid urease test. The rapid urease test was done by obtaining an inoculum, with a cotton swab, from Columbia agar (OXOID, Basingstoke, UK) cultures and transferring it to 0.5 mL Eppendorf tubes containing 0.4 mL Urea agar base (OXOID, Basingstoke, United Kingdom). In the case of C. *albicans*, its characteristic morphology was verified in wet mounts, and its purity was verified by Gram staining. 

### 4.2. Growth Curves of H. pylori J99 and C. albicans ATCC 10231 Strains

This assay was done to identify the timespan required to obtain *H. pylori* and *C. albicans* growing in their exponential phase in order to obtain the largest number of cells possible of each microorganism.

From a 36 h culture of *H. pylori*, whose morphology was verified after Gram staining, as described above, an initial inoculum containing 1.5 × 10^6^ colony forming units (CFU) mL^−1^ in 5 mL of Brucella broth (BB) (Difco, Wokingham, UK) supplemented with 10% fetal calf serum (Becton Dickinson, East Rutherford, NJ, USA) (BB-FCS) was prepared in 16 × 100 mm glass tubes (Pyrex, Swedesboro, NJ, USA). Cultures were incubated for 60 h at 37 °C under microaerobic conditions using a 3429 model incubator (Thermo Scientific, Waltham, MA, USA). Aliquots of 200 µL were obtained at 12 h intervals and transferred to wells of a 96 wells microplate, (TR5003 model, Thomas Scientific, Swedesboro, NJ, USA), and the turbidity was measured using an Infinite M200 PRO model spectrophotometer (TECAN, Männedorf, Switzerland) at 600 nm.

From a 24 h culture of *C. albicans,* an inoculum containing 1.5 × 10^6^ CFU mL^−1^ in 5 mL of BB-FCS was prepared and incubated at 37 °C during 12 h under aerobic conditions. Then, 200 μL of this suspension were incubated in a 96-well microplate (TR5003 model, Thomas Scientific, Swedesboro, NJ, USA), and the growth was evaluated measuring the absorbance at 600 nm every 4 h during 48 h using a model Infinite M200 PRO spectrophotometer (TECAN, Männedorf, Switzerland) following the procedure described by Maidan et al. [31]. In addition, the growth and cell morphology of yeasts were monitored by optical microscopy.

### 4.3. Preparation of Antibiotics Stock Solutions 

Stock solutions of AMX and CLT (both obtained from Sigma Aldrich, St. Louis, MO, USA) were prepared in 1 M ammonia (Santa Cruz, Dallas, TX, USA) or M dimethyl sulfoxide (DMSO) (Sigma-Aldrich, St. Louis, MO, USA), respectively. Stock solutions were filtered using 0.22 μm pore size sterile filters (Thermo Scientific, Waltham, MA, USA) and then aliquoted at final concentrations of 0.24, 0.12, 0.06, 0.03 and 0.015 μg mL^−1^.

### 4.4. Determination of the MIC of Amoxicillin and Clarithromycin for H. pylori J99 

The CLSI agar dilution technique methodology [26] was used to determine the MIC of both antibiotics for *H. pylori* J99 using 6-well microplates (TR5003 model, Thomas Scientific, Swedesboro, NJ, USA) containing Müller Hinton (MH) agar (Difco, Wokingham, United Kingdom) supplemented with 7% horse serum (Biological Industries, Cromwell, CT, USA) plus the different antibiotics concentrations. A McFarland 2 initial inoculum was prepared from a *H. pylori* J99 strain cultured on Columbia agar for 72 h, and 3 μL of this inoculum were seeded in each well and incubated for 48 h. MIC was determined as the antibiotic concentration at which no bacterial growth was observed. 

### 4.5. Determination of H. pylori J99 Internalization into C. albicans ATCC 10231 Cells in the Presence of Amoxicillin or Clarithromycin

The subinhibitory concentration of AMX or CLT used in this assay was ¼ of the MIC determined by the previous experiment. From cultures of *H. pylori* J99 and *C. albicans* ATCC 10231 grown in plates until exponential phase (20 and 10 h, respectively), suspensions were independently prepared in 5 mL of BB-FCS. Then, the 5 mL of *C. albicans* suspension was added to the *H. pylori* suspension totaling 10 mL, and a final concentration of 3 × 10^8^ CFU mL^−1^ comparing the turbidity of the suspension with a MacFarland scale. Before incubating, either of the antibiotics (AMX or CLT) was added to reach a concentration equal to ¼ MIC. Then, cultures were incubated (Thermo Scientific, Waltham, MA, USA) at 37 °C under microaerobic condition, 10% CO_2_, and 20 µL of the co-culture was collected at times 0, 1, 3, 6, 12, 24 or 48 h. At each of these times, a sample was placed on a slide, covered with a coverslip and observed under an optical microscope (Leica, Wetzlar, Germany) using the 100X objective lens to determine the number of yeast cells containing BLBs. The percentage of yeast cells containing BLBs was calculated after counting yeast cells containing BLBs or not containing them in 100 microscope fields for each time sampled. A similar co-culture but lacking antibiotics was used as control. This assay—including co-cultures plus AMX, co-cultures plus CLT and control co-cultures without antibiotics—was done in duplicate. 

Then, the percentages of yeast cells containing BLBs in the presence or absence of antibiotics were incorporated into an Excel Microsoft 16.40 database (Microsoft, Redmond, WA, USA) and processed using the SPSS 23.0 software (IBM Company, Armonk, NY, USA). The levels of the categorical variables were expressed by incubation time and their percentages of BLBs containing yeasts, being the relationship between the categorical variables determined using the Student’s *t*-test. Values of *p* < 0.05 were considered significant.

The following assays were required to obtain yeast cells from the co-cultures free of extracellular *H. pylori* cells. For this purpose, inocula were obtained from the 48 h co-cultures described above using cotton swabs (QIAGEN, Hilden, Germany) and seeded on the surface of plates containing Sabouraud agar (Merck, Darmstadt, Germany) supplemented with chloramphenicol, following the instructions of the manufacturer (OXOID, Basingstoke, UK). Plates were incubated at 37 °C forz 48 h under aerobic conditions. After incubation, Gram stainings were prepared to confirm the absence of extracellular bacteria to continue with the following experiments (i.e., the cell viability assay and the extraction of DNA from yeasts).

### 4.6. Cell Viability Assay 

From the co-cultures showing the presence of yeasts harboring BLBs, 1 mL suspensions with a turbidity of 0.5 McFarland were prepared in sterile saline solution and 1 µL of the working solution of the LIVE/DEAD BacLight Bacterial Viability kit L-7012 (Thermo Scientific, Waltham, MA, USA) was added. After 15 min of incubation at room temperature in the darkness, suspensions were gently vortexed for 3 s. Then, 10 µL of each suspension was placed on a slide to be observed using the 100X objective lens of a fluorescence microscope equipped with an integrated camera (Motic, Viking Way, Richmond, VA, Canada). Filters used to evaluate cell viability were FITC (AT480/535) and TRITC (AT540/605) to observe the green fluorescence of SYTO 9 fluorochrome and the red fluorescence of propidium iodide, respectively. Images were processed using the ImageJ software (NIH Image, Bethesda, MD, USA) to merge the images obtained.

### 4.7. Detection of H. pylori Genes in Total DNA Extracted from C. albicans Cells

To avoid the contamination of yeast cells DNA with DNA from extracellular *H. pylori* cells, *C. albicans* cells previously co-cultured with the pathogenic bacterium having BLBs were seeded on Sabouraud agar supplemented with chloramphenicol and reseeded six times in this same medium to eliminate extracellular *H. pylori* cells from the cultures, a condition confirmed by Gram staining.

### 4.8. Extraction and Quantification of DNA from Yeasts

DNA was extracted from yeasts previously co-incubated with *H. pylori* cells and then incubated in chloramphenicol-supplemented Sabouraud agar (Merck, Darmstadt, Germany) to obtain yeast cells free of extracellular *H. pylori.* DNA was also extracted from pure cultures of *H. pylori* J99 (positive control) and *C. albicans* ATCC 10231 (negative control). In all cases, DNA was extracted using the UltraClean Microbial DNA Isolation kit (MO BIO, QIAGEN, Hilden, Germany) following the instructions of the manufacturer. To quantify the extracted DNA, 2 µL of each sample were placed in independent wells of a NanoQuant plate (TECAN, Männedorf, Switzerland), and the absorbance was measured at 260 and 280 nm using an Infinite M200Pro model spectrophotometer (TECAN, Männedorf, Switzerland). Samples having a 260/280 absorbance ratio between 1.8 and 2.0 were selected. These samples were kept at −20 °C until further use. Table 1 shows the quantification and purity of the DNA extracted from yeasts obtained from co-cultures in the presence of AMX or CLT and from the control cultured in the absence of antibiotics.

### 4.9. Amplification of the 16S rDNA Gene of H. pylori J99

The amplification of a 110 bp fragment of the 16S rDNA gene of *H. pylori* J99 was done by PCR using the SapphireAmp Fast PCR Master Mix kit (TAKARA BIO INC, Shiga, Japan). For this purpose, 12.5 μL master mix, 1 μL forward primer (Table 2), 1 μL reverse primer (Table 2), 2 μL of sample or control DNA and 8.5 μL PCR grade H_2_O were used, totaling 25 μL of PCR mixture. PCR conditions were: initial denaturation at 94 °C for 1 min, denaturation at 98 °C for 5 s, hybridization at 53 °C for 5 s and extension at 72 °C for 40 s. All samples were subjected to 30 PCR cycles using a thermocycler (Fermelo Biotec T 960, Providencia, Santiago, Chile).

### 4.10. Agarose gel Electrophoresis 

After amplifying the 110 bp fragment of *H. pylori* J99 16S rDNA gene, samples were subsequently analyzed by 2% agarose gel electrophoresis. Agarose (0.6 g) (Lonza, Walkersville, MD, USA) was dissolved in 30 mL of 1X TAE (Thermo Scientific, Waltham, MA, USA), and then 0.6 μL GelRed (Biotium, Landing Parkway, Fremont, CA, USA) were added. Ten microliter of the products of amplification and 1 μL of molecular weight marker (100 bp marker (MAESTROGEN, Hsinchu, Taiwan)) were loaded in the gels. Electrophoresis was run for 90 min at 70 V. Bands were observed and recorded under an Enduro model transilluminator (Labnet, Edison, NJ, USA)

## 5. Conclusions

The internalization of *H. pylori* to *C. albicans* in an environment in the presence of an antibiotic is dependent on the type of antibiotic used, suggesting that a therapy that includes amoxicillin could stimulate the entry of the bacterium to *Candida*, thus adversely affecting the success of treatment.

## Figures and Tables

**Figure 1 pathogens-10-00382-f001:**
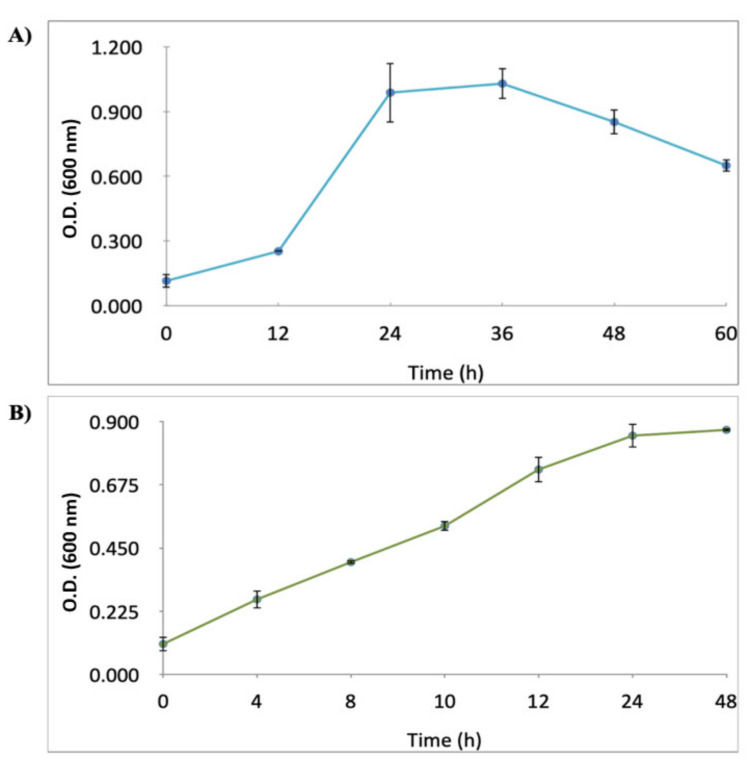
Growth curves of *H. pylori* J99 (**A**) and *C. albicans* ATCC 10231 (**B**) cultured in Brucella broth fetal calf serum under microaerobic conditions. O.D.: optical density. Mean of triplicates.

**Figure 2 pathogens-10-00382-f002:**
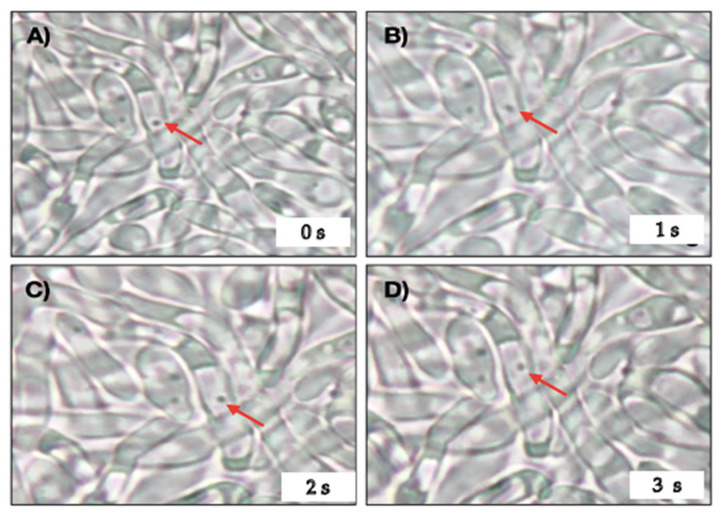
Optical microscopy of wet mounts of a 24 h *H. pylori* J99-*C. albicans* ATCC 10231 co-culture in the presence of a subinhibitory concentration of AMX. Red arrows indicate the movement of a bacteria-like body (BLB) within a yeast cell in micrographs taken at intervals of 1 s. AMX: amoxicillin. (**A**) Shows the initial position of a BLB at time zero (0 s) and (**B**–**D**), show in the change in position of the BLB at 1 s intervals. Appendix A Movement of BLBs within vacuoles.

**Figure 3 pathogens-10-00382-f003:**
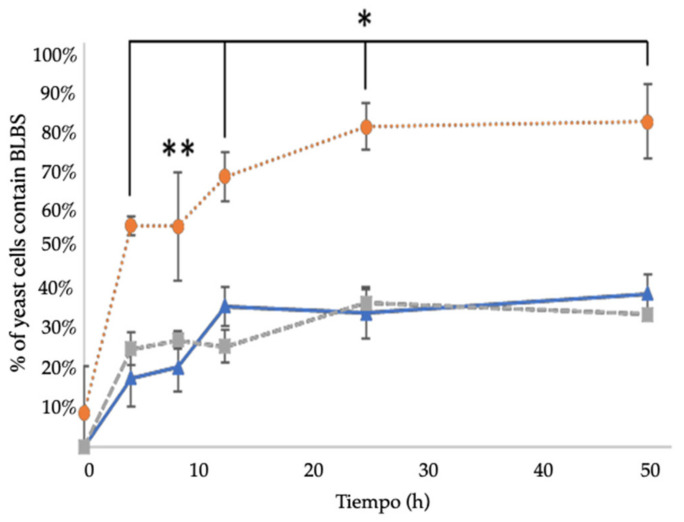
Percentage of *C. albicans* yeast cells containing BLBs, putatively *H. pylori*, evaluated by optical microscopy after co-culturing *C. albicans* ATCC 10231 and *H. pylori* J99. Co-culture in the absence of antibiotic (control) (
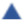
), co-culture in the presence of ¼ MIC CLT (
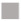
), co-culture in the presence of ¼ MIC AMX (
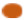
). Mean of duplicates. * *p* < 0.05, ** *p* < 0.1 with respect to control. BLBs: bacteria-like body MIC: minimal inhibitory concentration, CLT: clarithromycin, AMX: amoxicillin.

**Figure 4 pathogens-10-00382-f004:**
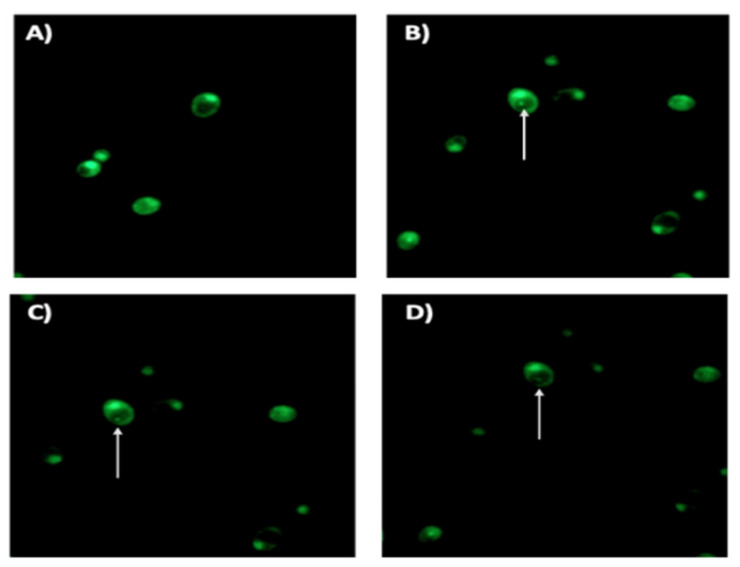
Fluorescence microscopy of a 48 h *H. pylori* J99-*C. albicans* ATCC 10231 co-culture in the presence of ¼ MIC AMX showing yeast cells containing BLBs, putatively *H. pylori*. (**A**) shows a pure culture of *C. albicans* 10231 (negative control) while (**B**–**D**) show micrographs of a viable BLB (arrow) within a viable yeast cell taken at 1 s intervals. AMX: amoxicillin, BLBs: bacteria-like body.

**Figure 5 pathogens-10-00382-f005:**
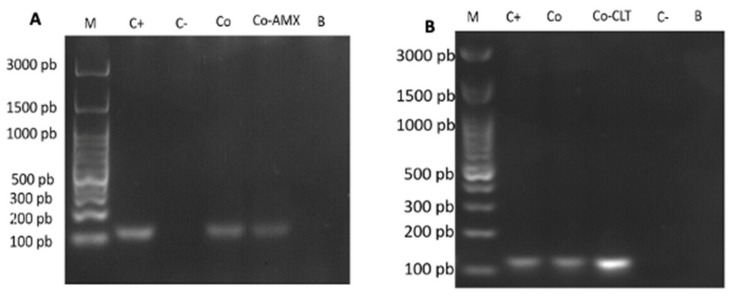
Images from 2% agarose gel electrophoresis showing the amplicons obtained after PCR amplification of the 16S rDNA gene of *H. pylori* in total *C. albicans* DNA after incubating an *H. pylori* J99-*C. albicans* ATCC 10231 co-culture during 48 h in the presence of AMX (**A**) or CLT (**B**) and then incubating on chloramphenicol supplemented Sabouraud agar to eliminate extracellular bacteria. Expected size for the amplicon:110 bp. M: molecular weight marker; C+: positive control (pure *H. pylori*); C-: negative control (pure *C. albicans*), Co: co-culture without the antibiotic; Co-AMX or Co-CLT: co-culture in the presence of ¼ MIC AMX (**A**) or ¼ MIC CLT (**B**); B: blank (PCR grade water). AMX: amoxicillin, CLT: clarithromycin.

**Table 1 pathogens-10-00382-t001:** Quantification and purity of DNA extracted from 48 h *H. pylori* J99-*C. albicans* ATCC 90028 co-cultures in the presence of AMX or CLT antibiotics and from a control without antibiotic.

Sample	Media	DE	
Control	35.4	3.39	B
AMX	46.85	2.19	A
CLT	31.15	1.59	B

AMX: amoxicillin, CLT: clarithromycin. According to the Tukey statistical test, mains sharing the same letter are not significantly different (*p* > 0.05).

**Table 2 pathogens-10-00382-t002:** Primers used for the amplification of *H. pylori* J99 strain *16S rRNA.*

Gene	Sequence	Tm (°C)	bp (Amplicon)	Reference
*16S rDNA*	F-5′CTCGAGAGACTAAGCCCTCC3′ R-5′ATTACTGACGCTGATGTGC3′	53	110	[19,26,57,58,59,60]

## Data Availability

No comments.

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
