# Peer review of "Antibiotics as a Stressing Factor Triggering the Harboring of Helicobacter pylori J99 within Candida albicans ATCC10231"

_pathogens, 2021, doi:10.3390/pathogens10030382_

Round 1

Reviewer 1 Report

I can only say that, with these revisions taken into account, the manuscript is now suitable for publication.

Author Response

Dear, 

We appreciate your comments 

Reviewer 2 Report

Revised manuscript is considerably improved. 

Now, there are several typos yet.

The style of unit symbol is confused. There are both ml and mL. 

Check The " CO2" in page7, line 239. "2" is subscript.

Check style of "°C".

Check The "1.5 x106" in page7, line 265. "6" is superscript.

Author Response

Dear, 

We appreciate your comments 

Reviewer 3 Report

The paper is fascinating and novel.

  • Have you detected H.pylori by Immunofluorescence?
  • Have you tested only Hp J99?
  • I think it's necessary to test other strains of Hp and a negative control

Author Response

Dear, 

We appreciate your comments 

This manuscript is a resubmission of an earlier submission. The following is a list of the peer review reports and author responses from that submission.

Round 1

Reviewer 1 Report

This article by Kimberly Sanchez-Alonzo et al. describe very interesting data on Hp/CanAlb co-culture and internalization.

These results are very well-written but deserve some revision to be very complete.

Global:

  • all acronym have to be full exaplined when first used.
  • Part Patents is not justified please suppress.

Introduction :

  • Is CanAlbicans very frequent in gastric mycobiome? How do the authors consider its impact on antibiotic resistance? Do the authors would recommend to and antifungal molecules to overcome possible fungi/bacteria interaction ?

Results :

  • Why have the authors considered OD and not UFC/ml? It is well-known that OD is not that efficient for Hp quantification. Please complete.
  • Have the authors verified the MIC for Hp, as the strains were collected from non-ATCC laboratory with an unknown number of culture? Moreover, it could be very interesting to determine MIC of Hp coculture with CanAlb, could the authors determine if this MICs were different ?
  • Figure 1 and 3 : what is indicate in the boxplot (median?mean?)? the authors stated they do the experiments in duplicate only, why not in triplicate as usual?
  • Figure and Table title have to be indicated in bold police.
  • Figure 5 and 6 could be reunite to ease the understanding.

Methods : 

  • Have the authors determined if DNA quantification were different? If yes, it could be of interest to normalize DNA input before PCR. Moreover, authors have to justify the choice of the used primers, as PCR performances for Hp could be very different.
  • Why have the authors not performed their analysis using CLT-resistant strains, as they clearly indicate that a global increase of this resistance spread worldwide? These informations were important and have to be add to the manuscript.

Reviewer 2 Report

In this study, the authors demonstrated that amoxicillin could stimulate the entry of the bacterium to Candida, and the bacterial system may damage the treatment success of H. pylori eradication. The results are very interesting, but there are some problems, and much remediation is needed.

Following are some comments and questions on the manuscript:

Specific comments:

Authors evaluate the intracellular H. pylori within candida by optical microscopy in the results of Fig2 and 4. The results are uncertain and the method may be unsuitable. Authors should use immunohistochemical methods using anti-H. pylori antibody. In this connection, what is different among Fig4B, C, and D. These pictures seem to be same position of the same sample.

Authors evaluate the intracellular H. pylori within candida by PCR in the results of Fig5 and 6. The method may analyze non-living H. pylori. So, authors should count the colony of H. pylori by plate culture or perform RT-PCR.

Authors used only J99 strain of H. pylori. There are several genotypes in H. pylori strain. So, authors should investigate for other strains.

Minor comments:

There are so many typos in this article, such as italic of H. pylori, Superscript, the representation of the units. Authors should carefully revise these typos.

Authors should describe the percentage of oxygen when H. pylori was cultured under microaerobic condition.